# Typical Changes in Gait Biomechanics in Patients with Subacute Ischemic Stroke

**DOI:** 10.3390/diagnostics15050511

**Published:** 2025-02-20

**Authors:** Dmitry V. Skvortsov, Sergey N. Kaurkin, Natalya V. Grebenkina, Galina E. Ivanova

**Affiliations:** 1Center for Brain and Neurotechnology, Moscow 117513, Russia; 2Research and Clinical Centre, Moscow 107031, Russia

**Keywords:** post-stroke, hemiplegic gait, gait biomechanics, ischemic stroke, gait analysis

## Abstract

**Background/Objectives:** Gait dysfunction occurs in 80% of stroke survivors. It increases the risk of falls, reduces functional independence, and thus affects the quality of life. Therefore, it is very important to restore the gait function in post-stroke survivors. The purpose of this study was to investigate the functional changes of gait biomechanics in patients with hemiplegia in the subacute stage of ischemic stroke based on spatiotemporal, kinematic, and EMG parameters. **Methods:** Initial biomechanical gait analyses of 31 patients and 34 controls were selected. The obtained parameters were assessed and compared within and across the study groups (post-stroke hemiparetic patients and healthy controls) to determine the pathognomonic features of the hemiplegic gait. **Results:** The gait function asymmetry was characterized by reciprocal changes, i.e., harmonic sequences of gait cycles. The most significant changes were in the kinematics of the knee joint and the EMG activity in the anterior tibialis, gastrocnemius, and hamstring muscles on the paretic side. The movements in the lower extremity joints ranged from a typical amplitude decrease to an almost complete lack of movement or involuntary excessive movement, as can occur in the ankle joint. The knee joint showed two distinct patterns: a slight flexion throughout the entire gait cycle and knee hyperextension during the middle stance phase**. Conclusions:** The gait function asymmetry is characterized by reciprocal changes (in temporal gait parameters). The most significant changes included decreased amplitude in the knee joint and decreased amplitude of EMG of all muscles under study, except for the m. quadriceps femoris.

## 1. Introduction

Stroke is a massive healthcare issue, being one of the leading causes of disability and mortality in the world. About 87% of all strokes are ischemic (IS) [1]. According to Feigin et al. [2], 58% of all ISs occur in people under 70 years of age, and 11% in those aged 15 to 49 years.

One of the most common stroke-associated disabilities is hemiplegia, which is found in 88% of post-stroke patients [3]. According to Abdu et al. [4], hemiplegia is twice as frequent in IS survivors as in those with hemorrhagic stroke. Hemiplegia is associated with a variety of motor impairments, including gait dysfunction, which affects 80% of stroke survivors [5]. Clinically, the post-stroke walking disorder is characterized by altered muscle tone on the affected side, slower walking speed, muscle weakness and fatigue, postural imbalance, and, hence, gait asymmetry. All these increase the risk of falls and reduce functional independence, thus affecting the quality of life [6,7]. Therefore, it is very important to restore the gait function in post-stroke survivors.

Biomechanical gait parameters in stroke patients have previously been investigated [8,9,10,11]. People with stroke walk slower, with a shortened stride length, spending longer on the paretic side during the stance phase and limiting the joint range of motion on the paretic side. On the other hand, the stance phase is longer and the swing phase is shorter on the non-paretic side [8,9]. The kinematic parameters of the lower limb joints on the paretic side are characterized by a reduced range of motion [10]. Electromyographic examinations of lower extremity muscles in hemiplegic patients show changes in the electrical muscle activity manifested as absent or decreased amplitudes and premature or delayed muscle activity peaks [11].

In previous studies, several attempts were made to classify the post-stroke abnormal gait patterns in order to provide targeted treatments and rehabilitation strategies [12,13,14,15]. Dolatabadi et al. analyzed gait patterns of 68 stroke survivors using a pressure-sensitive walkway and identified three pathological gait groups [13]. Pauk et al. used a motion capture system and force plates to study the spatiotemporal gait parameters and joint moments in 41 patients and identified three gait groups as well [14]. Another study [16], based on full-scale spatiotemporal and kinematic data from motion capture, identified six gait groups in 36 hemiplegic stroke patients. In the literature analyzed, biomechanical gait analyses were performed on groups of patients including both ischemic and hemorrhagic stroke. In addition, some studies included patients in the acute, subacute, and chronic periods of stroke. This may have a significant impact on the variability of detectable gait changes in patients with hemiparesis.

The purpose of this study was to identify the typical pattern of gait biomechanics in patients with hemiplegia in the subacute stage of ischemic stroke based on spatiotemporal, kinematic, and EMG parameters.

The research hypothesis stated that the acute stage of ischemic stroke is characterized by a total decrease in the function of the paretic side.

## 2. Materials and Methods

This was a single-center retrospective observational study based on data from an ongoing database of patients with stroke recruited at the Center of Brain Research and Neurotechnologies (Moscow, Russia). Only IS survivors were included in the study to obtain a more homogeneous study population and avoid high intergroup variability [17]. All subjects provided written informed consent, approved by the local ethics committee (11/25-04-22, dated 25 April 2022).

Registration: ClinicalTrails.gov identifier: NCT06299943. Registered on 19 July 2021.

### 2.1. Study Population

A total of 170 initial biomechanical gait analyses were reviewed to select subjects for the study. Thirty one patients were included in the study (23 females, 8 males), 14 (45%) of them with right hemiparesis. The mean age of the included patients was 54.8 ± 10.1 years, and the mean post-stroke duration was 79.5 ± 42.6 days. The mean body mass index (BMI) was 27.3 ± 4.4 kg/m^2^. The mean lower limb muscle strength was grade 3.1 (by manual muscle testing), and the lower limb muscle tone of flexors and extensors of the hip and calf was grade 1–2 (by the modified Ashworth scale). The mean dynamic gait index value was 14.8. The mean Timed Up and Go test score was 28.3 s. All patients were able to stand and walk independently and did not use any assistive devices.

The study also included 34 healthy controls, with a mean age of 29.8 ± 7.9 years and a mean BMI of 20.6 ± 1.8 kg/m^2^.

Inclusion criteria: patients with a first episode of unilateral ischemic stroke in the subacute phase (< 180 days), aged 18–75 years, with sustained motor impairment in the form of unilateral hemiparesis, with no concomitant motor pathology.

Exclusion criteria: chronic stroke (> 180 days); hemorrhagic stroke; recurrent stroke; concomitant motor pathology; age over 75; close to normal parameters; recording errors. Parameters were considered close to normal if they did not exceed one standard deviation based on goniogram analysis. The final decision to include parameters in this group was made by an expert group of three independent specialists.

### 2.2. Measurements

The biomechanical gait analysis was performed using a Steadys system (Neurosoft, Ivanovo, Russia). Seven inertial sensors were attached to the subject’s pelvis, thighs, ankles, and feet (Figure 1). Each sensor recorded spatiotemporal and kinematic parameters of the gait, as well as functional EMGs of the lower limb muscles. We moved away from the SENIAM methodology of EMG electrode placement because we wanted to measure the muscle activity of the largest possible flexor and extensor muscle mass. Thus, the electrodes were placed across the hip in its middle third in the front and the back to register the activity of the quadriceps femoris and hamstring muscles, along the tibialis anterior and on both the gastrocnemius lateralis and the gastrocnemius medialis (further referred to as gastrocnemius) muscles (Figure 1). All electrodes were placed on each muscle at a distance of no more than 5 cm. Disposable surface electrodes (Mederen, Tel Aviv–Jaffa, Israel) were used.

EMG recording is influenced by many factors that can significantly reduce the quality of the information obtained [18,19]. These factors include individual muscle characteristics [20], the thickness of the subcutaneous fat [21], and the distance between the electrodes [22,23], but, with a slight difference in the distance and several other factors, this may not affect the signal magnitude [21]. The position of the electrodes on the muscle itself is also important [24], as well as many other factors. Depending on the objectives of the study, the electrode arrangement recommended by SENIAM [25] or other options are used [26,27]. In this study, we aimed to obtain an EMG envelope in the gait cycle for a subsequent analysis of possible changes in the on–off time of a large muscle mass, which is not dependent on the position of the electrodes [28].

All the participants wore shoes with heel counters and walked a 10-m distance back and forth, making turns at the end of the distance, until 30 gait cycles were recorded for each leg. The automatic algorithm discarded the beginning and ending steps, walking speed acceleration and deceleration, and any unsteady steps (loss of balance or stumbles). Only valid gait cycles were included in analysis [29]. As a result, a standard report was generated, which contained spatiotemporal parameters, sagittal kinematic data for the hip, knee, and ankle joints, and profiles of electrical activity from four muscles: the quadriceps femoris, the hamstring, the tibialis anterior, and the gastrocnemius.

### 2.3. Parameters

The assessed temporal parameters included gait cycle (GC) and individual time periods within the GC, measured as a percentage of the GC: stance phase (SP), single limb stance phase (SLS), double limb stance phase (DLS), and the beginning of the terminal double limb stance phase (BTDLS). The assessed spatial parameters included foot clearance, circumduction, velocity, and stride length. The recorded kinematic parameters included flexion and extension of lower limb joints evaluated from goniograms for each joint. The goniograms were calculated based on all valid GCs and were used to measure amplitudes (in degrees) and the corresponding phases (as a percentage of the GC) (Figure 2).

The hip joint evaluation included the amplitudes and phases of flexion at the beginning of the GC (Ha1 and Hx1, respectively), full extension (Ha2 and Hx2), and swing flexion (Ha3 and Hx3). The knee joint parameters included the amplitudes and phases of the first flexion (Ka1 and Kx1), the first extension (Ka2 and Kx2), and the second flexion (Ka3 and Kx3). The ankle joint was described using the amplitudes and phases of the first extension (Aa1 and Ax1), the first flexion (Aa2 and Ax2), the second extension (Aa3 and Ax3), and the second flexion (Aa4 and Ax4). The profiles of muscular bioelectric activity were used to determine the maximum amplitude (in microvolts) and its phase. The tibialis anterior showed two peaks of bioelectric activity (TAa1 and TAa2) with corresponding phases (TAx1 and TAx2). The gastrocnemius displayed one peak of the maximum amplitude (TSa) with a corresponding phase (TSx). The quadriceps femoris exhibited two peaks (QFa1 and QFa2) with their respective phases (QFx1 and QFx2). Finally, the hamstring demonstrated a single peak (HMa) with one phase (HMx).

### 2.4. Data and Statistical Analyses

Statistical analysis was completed using the Statistica 12.0 software. The Shapiro–Wilk test was applied to verify the normality of the distribution of quantitative parameters and showed that the distribution was not normal (*p* < 0.05). The intragroup comparisons were performed using the Wilcoxon test, while the intergroup ones used the Mann–Whitney U-test. A *p*-value of less than 0.05 was considered statistically significant.

## 3. Results

### 3.1. Comparison of Spatiotemporal Parameters

The following significant differences in spatiotemporal parameters were observed (Table 1). In the intragroup comparison, the paretic side demonstrated a shorter stance phase, an earlier start of the SLS and BTDLS phases, a lower clearance, and an increased circumduction compared to the non-paretic side (*p* < 0.05 for all).

The intergroup comparison demonstrated a longer GC, a DLS, and a lower clearance on both sides in the patient group versus the healthy control group (*p* < 0.05 for all). A longer SP, a delayed BTDLS phase, and a decreased clearance were observed on the non-paretic side. Earlier SLS and BTDLS phases and increased circumduction were found on the paretic side. Finally, velocity was significantly reduced in the patient group (*p* < 0.05 for all).

### 3.2. Comparison of Kinematic Parameters

Significant differences in the kinematic parameters were observed (Table 2). The hip joint on the paretic side presented decreased Ha1 and Ha3 and a decreased and delayed Ha2 (*p* < 0.05 for all). The knee joint on the same side showed a lower Ka1, a premature Ka2, and a lower and premature Ka3 (*p* < 0.05 for all). The amplitudes of the ankle joint on the paretic side were increased and premature for Aa1, decreased and premature for Aa2, premature for Aa3, and increased for Aa4 (*p* < 0.05 for all).

According to the intergroup comparative analysis, the hip joint on the paretic side had reduced and premature Ha1 and Ha2, and a decreased and delayed Ha3, whereas the non-paretic side showed a decreased and premature Ha1, a decreased and delayed Ha2, and a delayed Ha3 (*p* < 0.05 for all). The knee joint parameters on the paretic side presented a decrease in and early onset of Ka1 and a decrease in Ka2 and Ka3 (*p* < 0.05 for all), whereas the non-paretic side showed an early onset of Ka1 and a decrease and delay of Ka3 (*p* < 0.05 for all). The ankle joint on the paretic side demonstrated an increased premature Aa1, a decreased Aa2, a delayed Aa3, and an increased Aa4 (*p* < 0.05 for all), whereas the non-paretic side showed a premature Aa1, a delayed Aa2, a decreased and delayed Aa3, and an increased Aa4 (*p* < 0.05 for all).

### 3.3. Comparison of EMG Parameters

The following significant differences in EMG parameters were identified by intragroup comparative analysis (Table 2). The tibialis anterior of the paretic side showed a decreased TAa1 as well as a decreased and premature TAa2 (*p* < 0.05 for all). The paretic gastrocnemius was presented by a lower and premature TSa (*p* < 0.05). The quadriceps femoris on the paretic side had a lower QFa1, and its QFa2 appeared earlier than on the non-paretic side (*p* < 0.05 for all). The hamstring HMa peak was lower on the paretic side (*p* < 0.05).

As shown by the intergroup comparison, the paretic side presented a lower TAa1 and a lower and delayed TAa2 of the tibialis anterior (*p* < 0.05). The bioelectric activity of the non-paretic side showed a decreased and delayed TAa1 and a decreased and premature TAa2 (*p* < 0.05 for all). The gastrocnemius demonstrated a lower TSa on the paretic side and a delayed TSa on the non-paretic side (*p* < 0.05 for all). The quadriceps femoris showed an increased and delayed QFa1 and a decreased and premature QFa2 on the non-paretic side, and a delayed QFa1 and a lower and premature QFa2 on the paretic side (*p* < 0.05 for all). The HMa of the hamstring was reduced and premature on both sides (*p* < 0.05 for all).

The general joint function patterns on the non-paretic side were characterized by decreased amplitudes, with complete preservation of basic joint function patterns. On the paretic side, there was a significant reduction in amplitudes (up to 50% of the normal amplitude), with the basic movements preserved in the hip and knee joints only. In the ankle joint, the foot drop syndrome of varying severity was predominantly detected. These changes are clearly visible in Figure 3.

The muscle function varied more widely. The lower limb muscles on the non-paretic side, and even more so on the paretic side, showed a significantly reduced EMG activity. The non-paretic side was characterized by an above-normal EMG activity in the thigh muscles, while the paretic side demonstrated a near-normal activity of the quadriceps femoris and a significantly lower hamstring activity (Figure 4).

## 4. Discussion

The general gait parameters of the patient group showed typical changes in post-stroke biomechanics [30]. There was a slight asymmetry in the GC duration, a normal duration of the SP on the paretic side, a significant increase in the SP on the non-paretic side, and a decrease in the SLS phase on the paretic side.

The study showed an asymmetry characterized by a decreased clearance and an increased circumduction on the paretic side.

The velocity was 2.5 times lower in the patient group. Slow walking has a few well-known effects on general gait parameters [9,30,31]. Therefore, the increase in the DLS phase is partly related to the slower gait speed and hence the longer GC duration.

The differences in the BTDLS phase were important as they showed the reciprocity disorder. The non-paretic limb makes initial contact with the ground too early to unload the paretic limb. On the other hand, the paretic limb hits the ground late. Normally, a harmonic gait is represented by both legs beginning each GC immediately when the opposite leg reaches 50% of the GC.

A comparison of our data with those published in the literature revealed some differences. Since changes in hemiparetic gait biomechanics have several variants [8], the published results also vary across different sources. In a review by Mohan et al. [32], the typical changes in the post-stroke gait kinematics included a decrease in hip joint extension, a decrease in ankle joint flexion during the support period, decreased amplitudes of the hip and knee flexion and ankle extension during the swing phase, as well as a decreased knee joint extension in the late swing phase. One specific type of locomotion is represented by knee hyperextension which locks the knee joint during the middle SP. The sources cited in [32] were published no earlier than 1993 and describe some of the patterns of gait biomechanics found in our study.

Abnormal EMG activity was noted in [32,33,34]: an increased activity of the tibialis anterior, an early activation of the gastrocnemius, and a late activation of the vastus lateralis. These patterns were also observed in our study, along with other variations.

Our study included only those patients who could walk independently; therefore, the study group of patients had relatively mild functional impairments. Since the biomechanical parameters on the paretic side are reduced, several important universal phenomena, that have been previously described, can be observed [35]. Physiologically normal parameters are optimal for the gait function. Therefore, if there is a functional ability to compensate for motor loss, the gait biomechanics adapt so that the paretic side can operate in a close-to-normal mode. In this study, this was seen in the typical changes in the temporal structure of the GC: the SP duration on the paretic side was indistinguishable from normal. As a result of the virtually normal SP duration, the associated parameters (such as Hx2, Kx3, and Ax2) on the paretic side were also closer to normal values than those on the non-paretic side.

The movements in the lower extremity joints ranged from a typical amplitude decrease to a nearly complete lack of movement or involuntary excessive movement, as can occur in the ankle joint. The knee joints showed two distinct types of qualitative functional changes. The first was a slight flexion throughout the entire GC without going into neutral. The second was a knee hyperextension which locked the knee joint during the middle SP.

Our results are consistent with those of some previous studies [15,16,36]. However, our patient population was characterized by a more severe gait dysfunction manifested by an increased GC, a lower velocity, and decreased knee joint amplitudes. Additionally, we recorded EMGs from four flexor–extensor muscles of the knee and ankle joints, which allowed us to assess changes in muscle automaticity. With the exception of the quadriceps femoris, the other muscles under study showed a significant decrease in bioelectrical activity, as well as changes in automaticity, including earlier maximum activity (gastrocnemius) and additional activity in the middle of the GC (hamstring), or the main activity during the swing period (tibialis anterior).

Thus, the research hypothesis was largely confirmed. However, the variability of developing disorders proved to be quite high.

The results of the study allow us to better personalize the rehabilitation of patients in the subacute stage of stroke. The basis is formed by the biomechanical data from the gait study. The results of the study of muscle EMG suggest the possibility of using the method of multichannel functional electrical stimulation of muscles during walking. At the next stage, we plan to conduct a course of such stimulation for the muscles studied in this work.

## 5. Conclusions

In this study, the functional changes in biomechanical gait parameters typical for the subacute stage of stroke were identified.

The presence of functionally different types of gait disorders in the patient group increased the spread of data around the means. Further studies are needed to take into account the identified forms of gait pathology.

The gait function asymmetry is characterized by reciprocal changes, i.e., harmonic sequences of GCs. The most significant changes were in the kinematics of the knee joint and the EMG activity in the anterior tibialis, gastrocnemius, and hamstring muscles.

Future research could aim to identify accurate gait patterns in patients with subacute stroke. This information could be crucial in providing targeted rehabilitation in early stages of ischemic stroke.

## Figures and Tables

**Figure 1 diagnostics-15-00511-f001:**
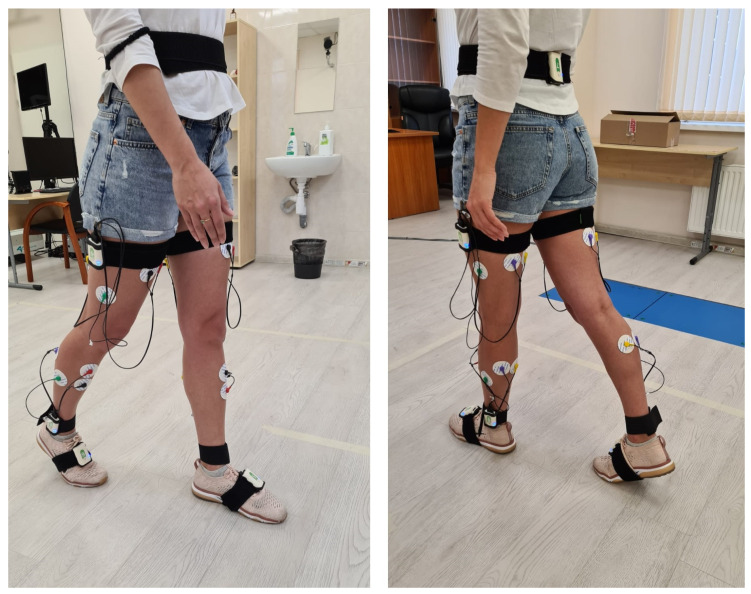
The process of recording gait parameters.

**Figure 2 diagnostics-15-00511-f002:**
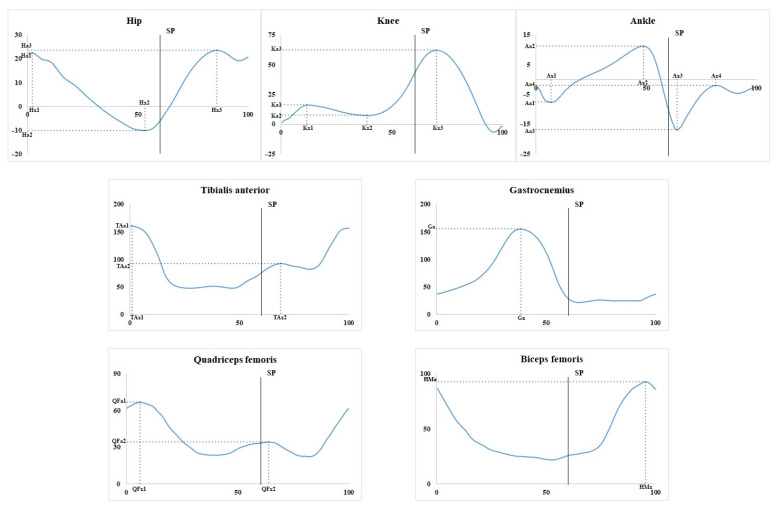
Joint goniograms and envelope EMGs with marked peaks of joint movements and electromyographic activity and their corresponding amplitudes and phases, respectively. The horizontal axis presents the GC in % and the vertical axis presents the amplitude in degrees (for joints), where zero degrees is the neutral position of a joint, positive and negative regions are flexion and extension, respectively, and electromyographic activity is provided in microvolts. SP marks the vertical line that represents the end of the stance phase.

**Figure 3 diagnostics-15-00511-f003:**
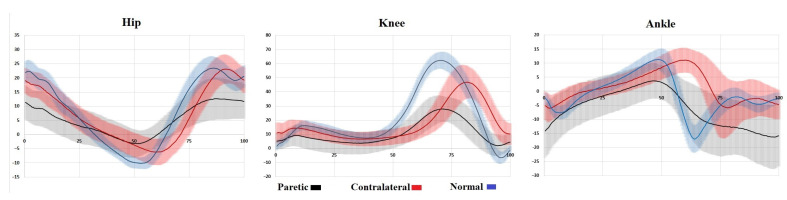
Joint goniograms for the paretic and non-paretic sides and the healthy controls. The vertical axis presents the amplitude in degrees, where zero degrees is the neutral position of a joint, positive and negative regions are flexion and extension, respectively, and the horizontal axis presents the GC in %. The solid lines represent the average values, and the light shading shows the respective standard deviations.

**Figure 4 diagnostics-15-00511-f004:**
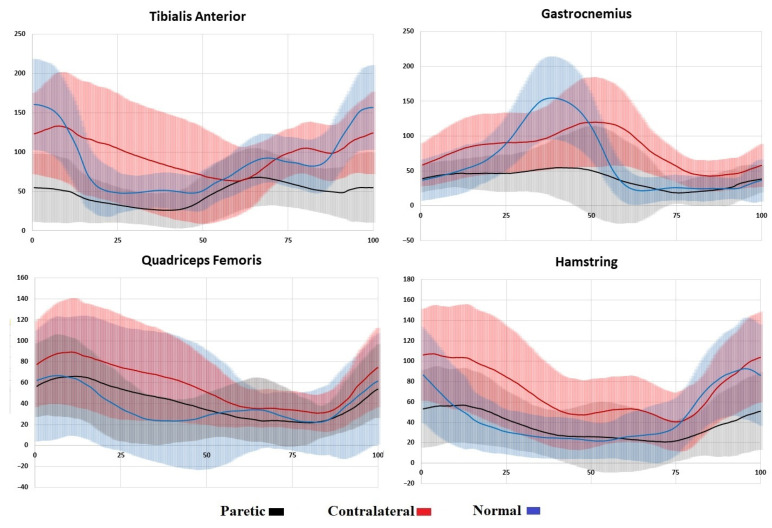
Envelope EMGs of the paretic and non-paretic sides and the healthy controls. The vertical axis presents the EMG amplitude in μV, and the horizontal axis shows the GC in %. The solid lines represent the average values, and the light shading represents the respective standard deviation.

**Table 1 diagnostics-15-00511-t001:** Spatiotemporal parameters of the patient and healthy control groups.

Spatiotemporal Parameters	Side	Patient Group (*n* = 31)	Healthy Control Group (*n* = 34)
GC (s)	paretic	1.5 [1.4; 1.7] *	1.1 [1.1; 1.2]
	non-paretic	1.6 [1.4; 1.7] *	
S (%)	paretic	63.2 [61.7; 67.7] ^#^	63.1 [62.4; 64.4]
	non-paretic	76.4 [79.3; 80.5] *	
SLS (%)	paretic	24.3 [20.2; 26.4] *^#^	36.9 [35.7; 37.9]
	non-paretic	36.8 [33.1; 38.7]	
DLS (%)	paretic	38.4 [36.1; 45.7] *	26.1 [24.6; 28.1]
	non-paretic	39.4 [36.2; 45.0] *	
BTDLS (%)	paretic	42.2 [40.0; 45.8] *^#^	49.9 [49.6; 50.3]
	non-paretic	57.4 [55.1; 59.8] *	
Clearance (cm)	paretic	9.0 [7.0; 13.0] *^#^	13.5 [12.0; 15.0]
	non-paretic	12.0 [10.0; 13.0] *	
Circumduction (cm)	paretic	4.0 [2.0; 6.0] *^#^	3.0 [2.0; 4.0]
	non-paretic	2.0 [2.0; 3.0] *	
Velocity (km/h)		1.67 [1.06; 1.96] *	4.27 [4.07; 4.5]
Stride length (cm)		72 [48; 81] *	137 [129; 147]

Note: *—significant difference from the healthy control, *p* < 0.05; ^#^—significant difference from the non-paretic side, *p* < 0.05.

**Table 2 diagnostics-15-00511-t002:** Joint and EMG parameters of the patient and healthy control groups.

Joints/Muscles	Parameter	Side	Patient Group (*n* = 31)	Healthy Control Group (*n* = 34)
Hip joint (°)	Ha1	paretic	11 [6; 18] *^#^	22.5 [20; 25]
		non-paretic	19 [16; 22] *	
	Hx1	paretic	1 [0.5; 4.7] *	2.2 [1.7; 2.7]
		non-paretic	0.7 [0.5; 3.5] *	
	Ha2	paretic	−3 [−7; 0] *^#^	−11 [−12; −9.5]
		non-paretic	−7 [−10; −3] *	
	Hx2	paretic	51.2 [47.2; 53.7] *^#^	52.6 [50.7; 55.3]
		non-paretic	61.5 [57.5; 63.2] *	
	Ha3	paretic	12 [8; 19] *^#^	24 [22; 27]
		non-paretic	23 [21; 27]	
	Hx3	paretic	91 [85.7; 95.7] *	86.8 [84.2; 89.3]
		non-paretic	91.7 [90.2; 92.7] *	
Knee joint (°)	K0	paretic	5 [0; 9] *^#^	2.5 [−1.5; 5]
		non-paretic	11 [4; 16] *	
	Ka1	paretic	9 [7; 14] *^#^	17 [14; 19]
		non-paretic	17 [12; 19]	
	Kx1	paretic	8.2 [6.4; 12.2] *	12.2 [11.6; 14]
		non-paretic	8.5 [6; 11.2] *	
	Ka2	paretic	1 [−3; 6] *	6 [4; 9]
		non-paretic	4 [2; 8]	
	Kx2	paretic	32.7 [29; 38.5] ^#^	37.4 [34.4; 40.5]
		non-paretic	38.7 [34.5; 42.7]	
	Ka3	paretic	32 [27; 37] *^#^	63 [60; 67]
		non-paretic	54 [48; 57] *	
	Kx3	paretic	70 [67.5; 75.2] ^#^	70.4 [69; 71.2]
		non-paretic	81.2 [78.5; 83.5] *	
Ankle joint (°)	A0	paretic	−13 [−19; −8] *^#^	−3 [−5; 0]
		non-paretic	−5 [−9; −2] *	
	Aa1	paretic	−13 [−19; −9] *^#^	−8 [−10; −6]
		non-paretic	−6 [−10; −3]	
	Ax1	paretic	0.75 [0.5; 2.2] *^#^	6.7 [5.5; 8]
		non-paretic	2 [1.2; 4.2] *	
	Aa2	paretic	5 [2; 10] *^#^	12 [10; 15]
		non-paretic	13 [11; 16]	
	Ax2	paretic	48 [42.7; 53] ^#^	48.2 [46; 50]
		non-paretic	59.5 [57.2; 65.5] *	
	Aa3	paretic	−16 [−27; −6]	−19 [−22; −15]
		non-paretic	−11 [−14; −6] *	
	Ax3	paretic	69.5 [66; 75.5] *^#^	64.2 [63; 65.2]
		non-paretic	77.2 [75; 82] *	
	Aa4	paretic	−12 [−23; −6] *^#^	−1 [−3; 1]
		non-paretic	−1 [−4; 2]	
	Ax4	paretic	86.5 [80.5; 95]	86.1 [81; 97]
		non-paretic	90.5 [86; 92]	
Tibialis anterior (µV)	TAa1	paretic	38 [28; 79] *^#^	158.5 [117.5; 186]
		non-paretic	117.5 [94; 169] *	
	TAx1	paretic	1.2 [0.5; 9.7]	1 [0.5; 2]
		non-paretic	7.2 [2; 10.2] *	
	TAa2	paretic	67 [46; 105] *^#^	154 [115.5; 184.5]
		non-paretic	97 [87.5; 142] *	
	TAx2	paretic	67.7 [65.5; 74] *^#^	99 [98.05; 99.7]
		non-paretic	80.7 [74.6; 83] *	
Gastrocnemius (µV)	TSa	paretic	51 [37; 68] *^#^	154 [112.5; 202]
		non-paretic	122 [98; 157]	
	TSx	paretic	34.5 [11; 41] ^#^	38.5 [36.6; 39.7]
		non-paretic	51.2 [46; 57] *	
Quadriceps femoris (µV)	QFa1	paretic	55 [34; 78] ^#^	62.5 [41; 86]
		non-paretic	78.5 [61; 106] *	
	QFx1	paretic	11.5 [9; 14.3] *	6.7 [4.5; 9]
		non-paretic	10.1 [7.3; 11.7] *	
	QFa2	paretic	24.5 [19; 29] *	56.5 [38.5; 81]
		non-paretic	38 [25; 48] *	
	QFx2	paretic	60.7 [52; 76.8] *^#^	99.7 [99.5; 100]
		non-paretic	72 [69; 73] *	
Hamstring (µV)	HMa	paretic	49 [37; 71] *^#^	83 [61.5; 123]
		non-paretic	106.5 [78; 154]	
	HMx	paretic	12.5 [6.2; 15.2] *	92.1 [43.1; 94.8]
		non-paretic	11.8 [1.2; 24.2] *	

Note: *—significant difference from the healthy control, *p* < 0.05; ^#^—significant difference from the non-paretic side, *p* < 0.05.

## Data Availability

The data used in this study are available at https://doi.org/10.17632/szmfkhyjpd.1, accessed on 31 May 2024.

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
