# Peer review of "Typical Changes in Gait Biomechanics in Patients with Subacute Ischemic Stroke"

_diagnostics, 2025, doi:10.3390/diagnostics15050511_

Round 1
Reviewer 1 Report
Comments and Suggestions for Authors
Thank you for the invitation to review the submission entitled, "Typical changes in gait biomechanics in patients with subacute ischemic stroke". The investigators present an interesting study that contains a cohort of 31 patients with subacute ischemic stroke and 34 controls examining the characteristics of gait biomechanics measured by inertial sensors and obtained electromyography. The authors conclude that "The obtained results may contribute to a better understanding of hemiplegic gait mechanisms and provide a tool for a more personalized and targeted rehabilitation strategy for patients with hemiplegia in early stages of ischemic stroke.". While this is an interesting result, I have several concerns that do not appear to be addressed within the current submission which I have listed below.
Introduction
Line 58-: The significance of this study was not clear. Please add the sentences regarding current issues and lack of evidence.
Methods
Line 71-88: Please add the information about walking aids and level of gait independence (Line 242 seemed to describe it).
Line 92-95: Please add the references of methodology of EMG electrode placement.
Line 103-107: Please add the detail information about gait cycle detection.
Line 111-138: Please add the references of the validity and accuracy of the IMU used in this study.
Results
Lines145-209: Please add the p-value (and/or effect size if possible) through the results section.
Figures and Tables: Please add the abbreviation.
Figure2: Please increase the font size of the data extraction point.
Figure 2 and 3: Please add any points where showed a significant difference.
Discussion
From the results of this study, I understand the differences from the control group. However, I did not know if the present results are specific features of patients with subacute ischemic stroke. Please clarify the characteristics and strengths of this study.
Author Response
Dear reviewer. Your questions and recommendations were very important not only for this article,
but also for continuing the work we started. Attached is a file with answers. The new version of
the article is also attached.
Thank you very much.

Reviewer 2 Report
Comments and Suggestions for Authors
The following are my comments.
This study investigated the functional changes of gait biomechanics in patients with hemiplegia in the subacute stage of ischemic stroke, using spatiotemporal, kinematic, and EMG parameters.
This article uses Steadys system (Neurosoft, Ivanovo, Russia) equipment to measure the gait in hemiplegia patients with the subacute stage of ischemic stroke, including EMG, kinematics and spatiotemporal parameters. They try to establish a typical gait pattern in hemiplegia patients with the subacute stage of ischemic stroke.
Here are a few questions and suggestions:
1. In general gait research, force plates are used to judge the stance and swing phases, as well as single-limb and double-limb support. Could you describe in detail how the instrument determines the stance phase, swing phase, single limb support, and double limb support?
2. Regarding Figure 2, here are a few suggestions to make it more straightforward.
(a) Is it possible to display the data of the experimental group and the control group at the same time in Figure 2? This allows readers to understand the differences between the two groups better.
(b) Is it possible to mark the boundary between the stance and swing phases in Figure 2?
(c) Figure 2 shows the maximum joint angle at the hip, knee and ankle, respectively, and the maximum amplitudes of EMG. However, the relationship among the gait cycle, maximum joint angle and maximum EMG amplitudes is not clearly stated. Is it possible to recreate a new figure so that readers can understand the relationship between the gait cycle, the maximum joint angle of the lower limbs, and the maximum EMG amplitudes of three muscles?
3. Regarding Table 1, here are a few questions to make it clearer
(a) Theoretically, Stance phase = SLS + DLS. Please explain the difference between how BTDLS and DLS are measured and calculated.
(b) Please explain how the Clearance is defined and measured in the article.
(c) Please explain how the Circumduction is defined and measured in the article.
(d) Is the step length missing in spatiotemporal parameters? If there is a measurement, can it be presented in table 1?
4. Regarding the presentation of muscle activity, it may be more meaningful to present the on and off times of muscle contraction. It may be possible to understand the difference in muscle activity timing between the two groups.
5. There must be an apparent asymmetry between the experimental and control groups regarding whether the figure and tables can show evident asymmetric phenomena.
Author Response
Dear Reviewer, Our team is very grateful to you for the effort you put into reviewing this article.
We have taken your recommendations into account and you will find many changes in the second version
of the article. We also attach a file with detailed answers to your questions.
With deep gratitude. The Authors.
